# Hepatic PTEN Signaling Regulates Systemic Metabolic Homeostasis through Hepatokines-Mediated Liver-to-Peripheral Organs Crosstalk

**DOI:** 10.3390/ijms23073959

**Published:** 2022-04-02

**Authors:** Flavien Berthou, Cyril Sobolewski, Daniel Abegg, Margot Fournier, Christine Maeder, Dobrochna Dolicka, Marta Correia de Sousa, Alexander Adibekian, Michelangelo Foti

**Affiliations:** 1Department of Cellular Physiology and Metabolism, Faculty of Medicine, University of Geneva, 1206 Geneva, Switzerland; flavienberthou@hotmail.com (F.B.); cyril.sobolewski@unige.ch (C.S.); margot.fournier@unige.ch (M.F.); christine.maeder@unige.ch (C.M.); dobrochna.dolicka@unige.ch (D.D.); marta.sousa@unige.ch (M.C.d.S.); 2Department of Chemistry, The Scripps Research Institute, Jupiter, FL 33458, USA; dabegg@scripps.edu (D.A.); aadibeki@scripps.edu (A.A.); 3Diabetes Center, Faculty of Medicine, University of Geneva, 1206 Geneva, Switzerland

**Keywords:** hepatokines, PTEN, FGF21, obesity, insulin resistance, NAFLD, liver, interorgan communication, metabolites

## Abstract

Liver-derived circulating factors deeply affect the metabolism of distal organs. Herein, we took advantage of the hepatocyte-specific PTEN knockout mice (LPTENKO), a model of hepatic steatosis associated with increased muscle insulin sensitivity and decreased adiposity, to identify potential secreted hepatic factors improving metabolic homeostasis. Our results indicated that protein factors, rather than specific metabolites, released by PTEN-deficient hepatocytes trigger an improved muscle insulin sensitivity and a decreased adiposity in LPTENKO. In this regard, a proteomic analysis of conditioned media from PTEN-deficient primary hepatocytes identified seven hepatokines whose expression/secretion was deregulated. Distinct expression patterns of these hepatokines were observed in hepatic tissues from human/mouse with NAFLD. The expression of specific factors was regulated by the PTEN/PI3K, PPAR or AMPK signaling pathways and/or modulated by classical antidiabetic drugs. Finally, loss-of-function studies identified FGF21 and the triad AHSG, ANGPTL4 and LECT2 as key regulators of insulin sensitivity in muscle cells and in adipocytes biogenesis, respectively. These data indicate that hepatic PTEN deficiency and steatosis alter the expression/secretion of hepatokines regulating insulin sensitivity in muscles and the lipid metabolism in adipose tissue. These hepatokines could represent potential therapeutic targets to treat obesity and insulin resistance.

## 1. Introduction

Non-alcoholic fatty liver disease (NAFLD) and insulin resistance (IR) are obesity-associated comorbidities representing major public health issues worldwide. Obesity is characterized by an abnormal and excessive fat accumulation in adipose tissue (AT) and development of ectopic fat depots in other peripheral organs (e.g., liver, muscle) mostly due to an imbalance between energy intake and expenditure. Lipotoxicity, associated with an abnormal accumulation of fat in these organs, generates a low-grade systemic inflammation and IR, affecting their metabolic homeostasis and leading to hyperglycemia [1,2,3]. If unresolved, chronic inflammation, lipotoxicity and glucotoxicity further impair pancreatic beta-cell functions and survival leading to the onset of type 2 diabetes (T2D).

Energy homeostasis in mammals is not only controlled by the central nervous system (CNS) but also by autocrine, paracrine and endocrine factors released by metabolically active peripheral organs [4]. This intercellular/inter-organ communication is mediated by a variety of different circulating factors including proteins/peptides (e.g., cytokines, hormones), metabolites (e.g., lipids, bile acids), nucleic acids (e.g., miRNAs) or nutrients (e.g., amino acids, vitamins) [5]. “Adipokines” (e.g., leptin, adiponectin), “hepatokines” (e.g., AHSG, FGF21) and “myokines” (e.g., irisin, myostatin) are defined as circulating endocrine factors exclusively, or predominantly, secreted by AT, the liver and muscles, respectively. Some of these molecules act directly on the CNS and/or on peripheral organs and contribute to the fine regulation of insulin sensitivity, glucose and lipid metabolism, and whole energy expenditure in pathophysiological contexts [4].

Several hepatokines were shown to worsen IR, inflammatory processes and metabolic homeostasis of AT, muscle, pancreas and/or kidney, and deregulation of their expression/secretion was reported to be closely associated with obesity, T2D and cardiovascular diseases [6,7]. Exemplifying such inter-organ metabolic regulation, the secretion of the hepatokine α-2-HS-glycoprotein (AHSG/fetuin-A) was shown to be increased in obese, insulin resistant and NAFLD patients [8] and to alter muscle insulin sensitivity [9,10] and adipocyte functions by interacting with Toll-like receptor 4 (TLR4) [11,12]. The leukocyte cell-derived chemotaxin 2 (LECT2) was reported to promote inflammation and IR in adipocytes and muscle cells via p38 [13] and JNK activation [14], respectively. Similarly, the expression/secretion of selenoprotein P (SELENOP), fibrinogen-like protein 1 (FGL1/hepassocin) and retinol binding protein 4 (RBP4) impaired insulin signaling and glucose metabolism in both hepatocytes and myotubes, thus contributing to IR and NAFLD development [15,16,17]. Finally, recent works have highlighted an increased expression/secretion of the hepatokine Tsukushi with obesity or cold exposure that was tightly associated with the presence of hepatic steatosis, inflammation and ER stress [18,19], but its role on body weight gain and glucose homeostasis still remains controversial [18,20].

Other specific hepatokines were, on the contrary, demonstrated to exert beneficial effects on metabolic homeostasis. The best characterized example is the liver-derived fibroblast growth factor 21 (FGF21), which enhances insulin sensitivity and glucose uptake in hepatocytes, adipocytes and muscle [21,22,23,24]. Additionally, FGF21 was also reported to increase energy expenditure by inducing the browning of white AT [25,26]. Other hepatokines such as bone morphogenetic protein (BMP) 4 and 7 [27,28,29], energy homeostasis associated (ENHO/adropin) [30] and irisin [31,32] improve insulin sensitivity and/or promote the browning of white AT. Finally, a novel hepatokine, apolipoprotein J, was recently characterized for its positive impact on muscle glucose metabolism and insulin sensitivity [33].

We previously reported that the hepatocyte-specific phosphatase and tensin homolog (PTEN) knockout (LPTENKO) mice exhibit an ambiguous phenotype. Indeed, despite developing a steatosis associated with hepatic IR, these mice are characterized by improved skeletal muscle insulin sensitivity and glucose tolerance, as well as a drastic reduction in white AT depots and the appearance of brown-like adipocytes in this tissue [34]. The link between hepatic PTEN silencing in the liver and the improved peripheral metabolism is unknown, but we hypothesized that the PTEN deficiency in hepatocytes triggers a positive inter-organ crosstalk between the liver and other metabolically active organs via secreted factors. The aim of this study was to identify liver-derived circulating factor(s), such as hepatokines, that are involved in this liver to muscle/AT crosstalk, which may represent novel biomarkers and potential therapeutic targets to restore peripheral insulin sensitivity and/or to decrease adiposity in obese and/or T2D patients. 

## 2. Results

### 2.1. Conditioned Medium from LPTENKO Primary Hepatocytes Improves Insulin Sensitivity in Muscle Cells and Inhibits Lipogenesis in Adipocytes

We previously reported that muscle insulin sensitivity and glucose uptake were increased, while adiposity was decreased in mice bearing a specific deletion of PTEN in hepatocytes (LPTENKO mice) [34]. To investigate whether the deletion of PTEN in hepatocytes induces the secretion of factors potentially promoting muscle insulin sensitivity and glucose uptake, in vitro differentiated C2C12 muscle cells were exposed for 24 h to conditioned media (CM) of mouse primary hepatocytes (MPH) isolated from 4-month old Pten^lox/lox^ (CTL) or AlbCre/Pten^lox/lox^ (LPTENKO) mice (Appendix A). Then, C2C12 cells were treated with insulin (10 nM, 10 min) and the phosphorylation of the insulin receptor and its major effector AKT were assessed by Western blot. Both basal and insulin-induced phosphorylation of insulin receptor and AKT were increased in C2C12 myotubes pre-treated with the CM of MPH from LPTENKO mice as compared to those from CTL mice (Figure 1A). Consistent with this gain of insulin sensitivity, a higher glucose uptake in response to insulin was observed in C2C12 myotubes pre-treated with the CM of MPH from LPTENKO mice (Figure 1B). This improved insulin sensitivity and glucose uptake in C2C12 myotubes was not associated with an upregulation of glucose transporters 1 and 4 (GLUT1 and GLUT4) or of insulin receptor substrates 1 and 2 (IRS1 and IRS2) (Appendix A), as also confirmed in vivo in the soleus muscle of LPTENKO mice compared to CTL mice (Appendix A). 

To determine whether the CM of MPH from LPTENKO mice could also regulate lipid accumulation in adipocytes, murine 3T3-L1 preadipocytes were differentiated in the presence of CM of MPH from CTL or LPTENKO mice (Appendix A). After 8 days of differentiation, lipid-droplet accumulation occurred normally in 3T3-L1 preadipocytes incubated with the CM of MPH from CTL mice, whereas this process was strongly inhibited (by ~80%) in cells treated with the CM of MPH from LPTENKO mice, as monitored by lipid droplets staining with BODIPY (Figure 1C). The absence of lipid droplets in 3T3-L1 preadipocytes incubated with the CM of MPH from LPTENKO mice was associated with a significant reduction, at both the gene and protein level, in major enzymes (ACC and FAS) and transcription factors (PPARG and SREBP-1C) promoting fatty acid biosynthesis (Figure 1D,E). Concomitantly, a strong downregulation of *Slc2a4* (GLUT4-coding gene) and *Gck* was also observed in these cells, indicating alterations of the glucose metabolism (Figure 1E). Consistent with this phenotype, the expression of fibroblastic and preadipocyte markers (e.g., *Cspg4*, *Fn1*, *Gli1* and *Pdgfrb*) increased in 3T3-L1 cells incubated with the CM of MPH from LPTENKO mice, whereas markers of mature adipocytes, e.g., *Adipoq* (adiponectin) and *Plin1* (a protein associated with the surface of lipid droplets), were downregulated (Figure 1E). Similar alterations of genes involved in lipogenesis and glucose metabolism (a decrease in *Acaca*, *Fasn*, *Gck*, *Plin1* and *Slc2a4* mRNA expression) were observed in vivo in the epididymal white AT (eWAT) of LPTENKO mice (Figure 1F).

Taken together, these data provide strong evidence that the deletion of PTEN in hepatocytes induces the secretion of hepatocyte-derived factor(s) promoting insulin sensitivity and glucose uptake in muscle cells, but inhibit differentiation, lipogenesis and lipid accumulation in adipocytes.

### 2.2. Protein Factors, but Not Metabolites, Secreted by PTEN-Deficient Hepatocytes Predominantly Trigger an Increased Insulin Sensitivity in Muscle Cells and Pre-Adipocytes Differentiation

Several classes of blood-circulating molecules are involved in inter-organs crosstalk including, for example, proteins/peptides, metabolites, nutrients or nucleic acids [5]. To delineate more precisely the nature of liver-derived secreted factors improving muscle insulin sensitivity and decreasing adiposity in LPTENKO mice, we first performed a metabolomic analysis of the plasma of CTL and LPTENKO mice. Analyses were focused on circulating bile acids, amino acids, biogenic amines, acylcarnitines, lysophosphatidylcholines, phosphatidylcholines, sphingomyelins and hexoses (Figure 2A). Of the 210 metabolites measured in these plasmas, 13 were significantly different (with an adjusted *p*-value < 0.15) between the CTL and LPTENKO mice, with six of them being increased (fold change KO/CTL ≥ 1.5), whereas seven were found to be reduced (fold change KO/CTL ≤ 0.67) (Figure 2B,C and Appendix A). To evaluate how metabolic alterations occurring specifically in the liver of LPTENKO mice may contribute to the observed alterations in the plasma metabolome, MetaboAnalyst algorithms were used to integrate metabolomic data with changes in the hepatic proteome of LPTENKO mice that we previously reported [35]. This integrative analysis indicates that among all the deregulated plasma metabolites, only the production of phosphatidylcholines was associated with two of the nineteen metabolic processes significantly altered in the liver of LPTENKO mice (FDR < 0.05) (Figure 2D and Appendix A). These analyses suggested that an altered production of phosphatidylcholines and derived metabolites by the liver of LPTENKO mice could potentially contribute to the increased insulin sensitivity and decreased adiposity observed in these mice. 

We next explored if protein factors secreted by the liver of LPTENKO mice were also involved in the metabolic improvements observed in the muscles and adipose tissues of these mice. To this end, in vitro experiments described in Figure 1 were repeated but in conditions where proteins in the CM of MPH were denatured by heating (90 °C, 30 min) prior to incubation with C2C12 myotubes or 3T3L1 cells. Under these conditions, the previously observed increase in insulin signalling mediated by CM of MPH from LPTENKO mice (Figure 1A) was totally blunted, thereby indicating that the integrity of circulating proteins within CM was required for an enhanced insulin responsiveness in C2C12 myotubes (Figure 2E). The same approach to investigate the impact of protein denaturation on the lipid and glucose metabolism of 3T3-L1 cells showed that heat-induced protein denaturation triggered downregulation of genes involved in lipogenesis and glucose metabolism in cells exposed to the CM of MPH from CTL mice. However, major differences in the expression of several lipogenic genes (e.g., *Acaca, Fasn, Pparg*) were abrogated in cells treated with heat-inactivated CM of MPH from CTL or LPTENKO mice (Figure 2F).

Together, these data indicate that, although we cannot totally exclude an impact of changes in metabolites secreted by PTEN-deficient hepatocytes, the improved insulin sensitivity and decreased adiposity in LPTENKO mice were mostly associated with protein factors secreted by hepatocytes of LPTENKO mice. 

### 2.3. PTEN Deficiency in Hepatocytes Alters the Expression of a Whole Network of Hepatokines

To uncover secreted hepatocytes-specific protein factors potentially triggering insulin hypersensitivity in muscles and decreased adiposity, CMs of MPH isolated from either CTL or LPTENKO mice were submitted to liquid chromatography and mass spectrometry (LC-MS/MS) analyses. Over the 780 identified proteins, 291 were deregulated (DEGs) in the CM of MPH from LPTENKO mice as compared to CTL mice. Among these 291 secreted proteins, 188 were upregulated (fold change CTL/KO ≤ 0.25) and 103 downregulated (fold change CTL/KO ≥ 1.33) in the CM of MPH from LPTENKO mice (Figure 3A and Appendix A). To narrow down the number of candidates of potential interest identified by our proteomic analysis, we then focused on well-established secreted proteins based on the presence of a signal peptide for secretion. Among the DEGs, only 80 proteins possess a signal peptide for secretion based on cross-referring with a list of predicted human secreted proteins established from three different prediction algorithms (SignalP 4.0, Phobius and SPOCTOPUS) and only 35 of the 80 DEGs were significantly deregulated based on an adjusted *p*-value < 0.15 (Figure 3B and Appendix A). MetaCore analyses of these 35 DEGs candidates indicated that 48.6% of them were associated with IR/T2D, 48.6% with the glucose metabolism, 37.1% with obesity and 25.7% with the lipid metabolism (Figure 3B). We thus further restricted our analyses to the 20 candidates involved in these processes with the addition of two other relevant factors, LECT2 and FETUB, which we identified in our proteomic analysis but, unexpectedly, were not identified by the MetaCore program, despite previous reports highlighthing their role in metabolic diseases [13,14,36,37] (Figure 3C). All of these 22 candidates (with the exception of CCL2, CTSL, RPS27 and USP14) are detected in human serum and are produced and secreted mostly, otherwise exclusively, by hepatocytes (Appendix A). An analysis of these 22 proteins with STRING database indicates that they constitute a network via protein–protein interactions and/or co-expressions patterns indicating a functional link in metabolic disorders (Figure 3D). Among these 22 candidates, the deregulation of seven hepatokines, i.e., AHSG (fetuin-A), ANGPTL4, FETUB (fetuin-B), FGF21, IGFBP1, IGFBP2 and LECT2, was further considered since, based on their role described in the literature and correlation of their expression with muscle insulin sensitivity, glucose tolerance and adiposity, they were potential candidate secreted factors associated with the metabolic phenotype of LPTENKO mice (Figure 3E and Appendix A). Of note, among these seven hepatokines, the FGF21 protein expression was detected only in CM of MPH from LPTENKO mice, whereas the ANGPTL4 protein was present only in CM of MPH from CTL mice. The remaining five hepatokines (AHSG, FETUB, IGFBP1, IGFBP2 and LECT2) were all decreased at the protein level in the CM of MPH from LPTENKO mice as compared to CTL (Figure 3C). Importantly, an mRNA analyses of these seven hepatokines in the liver of 4-month old LPTENKO mice correlated with the protein levels quantitated by our proteomic analysis, except for *Angptl4* and *Lect2*, indicating that the expression of AHSG, FETUB, FGF21, IGFBP1 and IGFBP2 was deregulated at the transcriptional level by PTEN deficiency in hepatocytes, whereas post-transcriptional mechanisms were likely involved in the deregulation of ANGPTL4 and LECT2 (Figure 3F). To exclude the idea that these alterations were due to adaptive and/or developmental compensatory mechanisms potentially occurring in the constitutive liver-specific PTEN knockout mice (LPTENKO mice), the deregulated mRNA expression of these seven hepatokines was further confirmed in a model of tamoxifen-inducible PTEN knockout specifically in hepatocytes (LIPTENKO) that we previously generated [38] at 5 months of age following the induction of the knockout at 2 months of age (Appendix A). Of note, three months after tamoxifen-induced PTEN silencing, LIPTENKO mice faithfully recapitulate the phenotype of LPTENKO mice with a constitutive deletion of PTEN in hepatocytes. These mice indeed display hepatomegaly, an improved glucose tolerance with a decrease glucose hepatic output in response to insulin, hepatic steatosis and a decreased adiposity (Appendix A).

Together, these data indicate that PTEN deletion in hepatocytes and/or the resulting hepatic phenotypes (e.g., IR and steatosis) triggers the alteration of the expression of a whole network of hepatokines, including AHSG, ANGPTL4, FETUB, FGF21, IGFBP1, IGFBP2 and LECT2, which likely affect insulin sensitivity and/or the lipid/glucose metabolism in distal peripheral organs.

### 2.4. Hepatokines Expression in Human and Mouse Models of Obesity, Hepatic Steatosis and Insulin Resistance

Although both humans and mice with obesity-associated hepatic steatosis display PTEN downregulation in the liver [39,40] and usually IR [41], the total abrogation of PTEN in mouse livers, on the contrary, triggers muscle insulin hypersensitivity and decreased adiposity despite the presence of a large steatosis [34]. Whether the expression of hepatokines associated with the phenotype of LPTENKO mice displays an opposite regulation in other mouse models of hepatic steatosis associated with obesity and IR is unclear. We therefore investigated the expression of the seven hepatokines uncovered by our proteomic analysis in the liver of a genetic mouse models of obesity, steatosis and IR/T2D, i.e., ob/ob mice (mice deficient for leptin) (Figure 4A and Appendix A) and db/db mice (mice deficient for leptin receptor) (Figure 4B and Appendix A). The expression of the above-mentioned hepatokines was also assessed in mice fed for 18 weeks with a high fat-containing diet (60% kcal from fat) [42] (Figure 4C). Finally, in silico analyses were performed to evaluate the mRNA expression of these hepatokines in publicly available transcriptomic datasets (Gene Expression Omnibus (GEO) datasets) of hepatic tissues from mice fed a high fat-containing diet for different time periods (9 and 12 weeks), in steatotic MAT1A knockout mice (Appendix A), or in the livers of human cohorts of patients suffering from obesity, steatosis, NASH and/or T2D (Appendix A). All these data are summarized in Appendix A to provide a general overview of the expression patterns of hepatokines in these different mouse models and human cohorts.

Altogether, these data revealed that the expression of *Igfbp1* and *Igfbp2* in other mouse models and patients with obesity-associated steatosis and IR is generally either unchanged or downregulated as in LPTENKO mice (Figure 4, Appendix A), thus suggesting that they may not be relevant in mediating the beneficial metabolic effects observed in LPTENKO mice. In contrast, while the expression of *Fetub* and *Lect2* ws found to be decreased in LPTENKO mice, these hepatokines are frequently upregulated in mouse models and patients with obesity, steatosis and IR (Figure 4, Appendix A), supporting a potential important role for these hepatokines in LPTENKO mice. Regarding *Fgf21*, this hepatokine was found to be almost always overexpressed in mice and humans with metabolic diseases (Figure 4, Appendix A), but such upregulation was previously demonstrated to result from the development of a resistant state to endogenous FGF21 action with obesity [43]. Therefore, *Fgf21* was also further considered as a potential driver of the metabolic improvements observed in LPTENKO mice. Finally, discrepant data between mouse models and patients were observed for the expression of *Ahsg* and *Angptl4*, for which we detected an opposite regulation of their mRNA expression with obesity, steatosis and IR (Figure 4, Appendix A). 

Together, these data suggest that the deregulated expression of *Ahsg*, *Angptl4*, *Fetub*, *Fgf21* and *Lect2* may be highly relevant in mediating muscle insulin hypersensitivity and decrease adiposity in LPTENKO mice, therefore highlighting these hepatokines as important modulators of insulin sensitivity, steatosis and obesity in both rodents and humans.

### 2.5. FGF21 Secreted by Primary Hepatocytes from LPTENKO Mice Triggers Insulin Hypersensitivity in C2C12 Myotubes

To assess the potential contribution of each of the seven candidate hepatokines in the insulin hypersensitivity of C2C12 myotubes and the differentiation and lipogenesis of 3T3-L1, we repeated the experiments described in Figure 1 with hepatokine-depleted CM of MPH isolated from either CTL or LPTENKO mice. To that end, MPH from CTL or LPTENKO mice were transfected with siRNAs specifically targeting each hepatokine and the CM from these cells were collected 48 h after MPH plating for further analyses (Figure 5A). 

Prior to the use of hepatokine-depleted CM, we first assessed that hepatokine expression was preserved as expected in 48 h cultured MPH from CTL and LPTENKO. As shown in Appendix A, 2 days post-plating (d2, 48 h) MPH were still well differentiated (*Alb* and *Serpina1a* differentiation markers are still expressed) and the expression of hepatokines of interest was still significant. Importantly, differences in the expression of specific hepatokines, as observed in liver tissues of CTL versus LPTENKO mice (Figure 3F) or in our proteomic analyses of CM collected 24 h post-plating (Figure 3C) were still preserved (Appendix A). Finally, each hepatokine was significantly silenced by transfection with specific siRNA in MPH 2 days after plating (Appendix A). Of note, although it appears that the impact of PTEN deficiency on the mRNA expression of specific hepatokines might be slightly attenuated after two days of MPH culture (Appendix A), hepatokines are secreted and accumulate to different extents in the CM of CTL versus LPTENKO MPH during the first 24 h of plating, as indicated by our proteomic analyses. Finally, although MPH are laborious to isolate and can be used only in a precise window of time (no later than 2–3 days post seeding), it was not possible to prepare functional CM from other currently used and classical in vitro cultured mouse or human transformed/immortalized hepatic cells. Indeed, all mouse and human cultured cells that we tested displayed an expression pattern of hepatokines totally different from mouse and human primary hepatocytes (Appendix A) associated with an important level of de-differentiation, as indicated by *Alb* and *Serpina1a* gene expression (Appendix A and Appendix A).

Since FGF21 is upregulated in MPH from LPTENKO hepatocytes, the depletion of *Fgf21* by specific siRNAs was performed in MPH from LPTENKO mice to mimic the FGF21 expression levels found in the CM of MPH from CTL mice (Figure 5A and Appendix A). On the contrary, all other hepatokines that were downregulated in MPH from LPTENKO mice were silenced by specific siRNAs in MPH from CTL mice in order to mimic the expression levels found in the CM of MPH from LPTENKO mice (Figure 5A and Appendix A). Furthermore, C2C12 myotubes were then exposed to CM that was depleted or not from these hepatokines prior to insulin stimulation and an evaluation of AKT phosphorylation by Western blotting was performed. As shown in Figure 5B, C2C12 myotubes incubated with CM of MPH isolated from LPTENKO but depleted of *Fgf21* by siRNAs displayed a significant reduction in insulin-induced AKT phosphorylation as compared with the unaltered CM of MPH from LPTENKO mice. In contrast, insulin-induced AKT phosphorylation in C2C12 myotubes was found to be unchanged when these cells were incubated prior to stimulation with CM of MPH from CTL mice with each of the other hepatokines (i.e., *Ahsg*, *Angptl4*, *Fetub*, *Igfbp1*, *Igfbp2* and *Lect2*) being downregulated (Figure 5B). To further confirm the role of FGF21 in C2C12 myotubes insulin hypersensitivity, we assessed insulin-induced AKT phosphorylation in C2C12 myotubes exposed to CM from CTL mice supplemented with increased concentrations of recombinant FGF21 for 24 h. As shown in Figure 5C, AKT phosphorylation in response to insulin was increased in a dose-dependent manner (trend at 100 ng/mL of FGF21 and significant from 500 ng/mL) in C2C12 myotubes by the addition of recombinant FGF21, thus confirming the responsiveness and sensitivity of C2C12 myotubes towards this hepatokine.

Together these data indicate that FGF21 upregulation, but not downregulation of the 6 others hepatokines, observed in the CM of MPH from LPTENKO mice is mostly responsible for the increased AKT phosphorylation induced by insulin in C2C12 myotubes. These data further suggest that liver-derived FGF21 production and secretion in vivo in LPTENKO mice strongly contribute to the increased muscle insulin sensitivity observed in these mice. 

### 2.6. AHSG, ANGPTL4 and LECT2 Are Required for 3T3-L1 Cells Differentiation and Lipogenesis

CM collected from MPH depleted of hepatokines were further incubated with 3T3-L1 to assess their effects on 3T3-L1 differentiation to pre-adipocytes and lipogenesis. As shown in the bright-field pictures, lipid-droplet (yellow bright dots) accumulation was reduced in 3T3-L1 cells differentiated only in the presence of CM of MPH isolated from CTL mice and with an siRNA-mediated downregulation of the expression of either *Ahsg*, *Angptl4* or *Lect2* (Figure 5D). These data were further supported by the mRNA expression of lipogenic/differentiation markers in these cells. Indeed, *Adipoq*, *Fasn*, *Plin1* and *Slc2a4* expression were found to be decreased in 3T3-L1 differentiated in the presence of CM of MPH from CTL mice silenced only for the three hepatokines, reaching similar expression levels as those observed in cells exposed to CM of MPH from LPTENKO mice (Figure 5E). 

These data indicate that these three hepatokines, i.e., AHSG, ANGPTL4 and LECT2, contribute to the efficient differentiation of 3T3-L1 cells towards pre-adipocytes and lipid accumulation in these cells. Based on these findings, it is likely that the decreased adiposity observed in LPTENKO mice is related, at least in part, to the downregulation of the expression and secretion of hepatic AHSG, ANGPTL4 and LECT2. 

### 2.7. Hepatokine Expression/Secretion in Hepatocytes Is Regulated by PTEN-Dependent Mechanisms and/or the Hepatic Metabolic Status

Whether PTEN deficiency or hepatic metabolic alterations induced by PTEN-deficiency trigger the deregulated expression and secretion of hepatokines by hepatocytes remains unclear. In order to disconnect potential effects related to PTEN downregulation versus lipotoxicity-associated with steatosis, we took advantage of mice with an inducible deletion of PTEN specifically in hepatocytes (LIPTENKO mice). the expression of hepatokines of interest was investigated in the liver of these mice one week after the induction of PTEN deletion by tamoxifen administration. In this short period, PTEN expression in the liver of these mice was found to be strongly reduced but steatosis had not yet developed (Figure 6A). One week after PTEN deletion, only *Ahsg*, *Fgf21* and *Igfbp1* expressions were altered similarly to in the hepatic tissues of constitutive liver-specific PTENKO mice (LPTENKO mice) and in CM of isolated MPH from LPTENKO mice (Figure 6A). This strictly PTEN-dependent regulation of the expression of these three hepatokines was further supported by the observed downregulation of *Ahsg* and *Igfbp1*, and upregulation of *Fgf21,* in isolated MPH from CTL mice transfected with siRNAs against *Pten* (Appendix A). Consistent with overactivation of the AKT signalling axis by PTEN deficiency, insulin triggered *Igfbp1* downregulation and *Fgf21* upregulation in addition to a slight but significant decrease in *Angptl4* expression in MPH isolated from CTL mice (Figure 6B). Of note, the exposure of MPH from CTL mice to a high glucose concentration also induced the gene expression of *Fgf21* but failed to significantly alter the expression of other hepatokines (Figure 6B).

Conversely, *Fgf21* expression was significantly reduced, whereas *Igfbp1* expression was strongly induced in MPH from LPTENKO mice when the PI3K/AKT signalling pathway was inhibited by specific inhibitors such as the LY294002 (Figure 6C). The inhibitor of the AMPK pathway (compound C) also tends to restrain *Fgf21* expression in MPH isolated from LPTENKO mice, whereas the inhibition of the AMPK pathway (compound C) promoted both *Igfbp2* and *Lect2* expression (Figure 6C). Of note, the expression of *Ahsg*, *Angptl4* and *Fetub* in MPH was almost insensitive to all other signalling pathway inhibitors (PI3K, MAPK, mTOR and AMPK pathways) tested here (Figure 6C). 

The impact of anti-diabetic drugs, some of which are still used in clinical settings, was further investigated with regards to their effects on the expression of our candidate hepatokines in MPH isolated from CTL mice. As shown in Figure 6D, PPAR activators such as rosiglitazone and the phytoestrogen genistein induced the mRNA expression of three insulin-sensitizer hepatokines (*Fgf21*, *Igfbp1* and *Igfbp2*) in MPH from CTL mice. Of interest, all the anti-diabetic compounds tested decreased the expression of *Ahsg* in MPH from CTL mice, which is consistent with the beneficial effect of inhibiting this hepatokine as highlighted by our data. These data indicate that classical anti-diabetic drugs may exert, at least in part, their beneficial effects on systemic IR and adiposity by modulating the expression and secretion of key hepatokines regulating these metabolic processes in peripheral organs. 

## 3. Discussion

Over the last years, intense efforts have been made to identify and target liver-derived factors in order to improve metabolic homeostasis in pathophysiological conditions such as obesity, IR and T2D. For this purpose, liver-specific PTEN knockout mice prove to be a useful model since these mice exhibit hepatic steatosis associated with hepatic IR, but also an improved skeletal muscle insulin sensitivity and a reduction and browning in white AT depots [34]. In this study, we provide evidence that a decreased PTEN expression/activity in the liver, as is observed in patients suffering from obesity-associated fatty liver disease [40], and the associated hepatic steatosis/IR, significantly modulate the pattern of hepatokines expression and secretion, resulting in an increased muscle insulin sensitivity and decreased adiposity in mice. We further identified a set of four hepatokines, whose expression and secretion are altered in these conditions, thereby significantly affecting the glucose and/or the lipid metabolism of muscle and adipose cells. 

Growing evidence indicates that metabolites such as lipids, amino acids and bile acids may trigger inter-organ communication between metabolically active organs, thereby contributing directly or indirectly to the pathogenesis of obesity, IR and T2D [44,45,46]. Our data identified phosphatidylcholines (PCs) as potential metabolites produced by the liver of LPTENKO mice which might contribute to improved IR and adiposity. PCs are the most abundant class of phospholipids and major component of cellular membranes. Specific members of this family (C12:0/C12:0 and C18:0/C18:1) were also suggested to improve hepatosteatosis, as well as muscle metabolism, glucose tolerance and IR in obese mice [47,48]. However, these conclusions were recently challenged by the fact that a high PC/PE (phosphatidylethanolamine) ratio in the liver is associated with IR [49,50], thus questioning these potential beneficial effects of PCs. The hepatic production of phosphatidylcholines appears to be significantly altered in the liver of LPTENKO mice and our data do not allow us to totally neglect the role of this class of metabolites in the improved insulin sensitivity and decreased adiposity observed in these mice. However, the thermo-sensitivity of CM from LPTENKO mice strongly supports instead the involvement of secreted protein factors in the observed metabolic improvement, leading us to focus our analyses on hepatokines. Further studies are however required to investigate in more depth the pathophysiological outcomes of modulating phosphatidylcholines in obesity-associated metabolic disorders. 

Identifying specific hepatic components secreted in the human or animal blood circulation and responsible for mediating liver to other peripheral organs crosstalk is extremely complex and challenging. The production and secretion of hormone-like protein factors can indeed originate from multiple organs and tissues, may have circadian regulation and the factors are usually present in a relatively low amount in the circulation. An unbiased high-throughput screening for such factors using omics approaches is further complicated by the presence of abundant plasma proteins (e.g., *albumin*), which mask the signal for low-expression circulating factors. We therefore developed an in vitro system that allowed us to detect protein factors secreted directly by primary hepatocytes and to then monitor, through siRNA-based protein targeting, the role of candidates of interest on muscle and adipose cells physiology. Using this approach, our data highlighted an altered synthesis and secretion by PTEN-deficient hepatocytes of FGF21, AHSG, ANGPTL4 or LECT2, which significantly contribute to improved insulin sensitivity in muscle cells and impaired lipid accumulation in adipose cells. Our rescue experiments using recombinant FGF21 protein further support the role of this hepatokine in the improved metabolic phenotype observed in LPTENKO/LIPTENKO mice. It is likely that other liver-derived factors also contribute to the PTEN-deficient liver crosstalk with other organs. Indeed, mass spectrometry analyses have a limited threshold in the detection of proteins; therefore, candidates identified and considered in our analyses were selected bioinformatically based on the presence of potential signal peptides for secretion as well as literature-based evidence. We cannot exclude that others factors secreted by different mechanisms (e.g., vesicular transport), not detected by mass spectrometry in the hepatocytes media, or with no previously reported evidence of a potential hormone-like role in regulating metabolism, may play a role in muscle insulin sensitivity and adiposity. Further studies are therefore required, specifically to investigate the metabolic role of factors with currently unknown functions, identified in our proteomic analyses. 

Among the seven hepatokines selected in our analyses and that display a synthesis and secretion significantly deregulated in PTEN-deficient hepatocytes, only the upregulation of FGF21 was able to foster insulin sensitivity of muscle cells by increasing insulin-induced AKT serine phosphorylation. We and others previously reported that hepatic gene expression and secretion of FGF21 was stimulated in LPTENKO mice [34,51]. Importantly, FGF21 is not only produced by the liver, but also by the AT and skeletal muscle [52] and the insulin-sensitizer role of FGF21 is well established in peripheral organs [22,24,53]. Our data show, in addition, that FGF21 produced by hepatocytes directly modulates muscle insulin responsiveness without the need for AT-derived adiponectin as a mediator for its action as was previously suggested [23,54]. The direct effects of FGF21 on skeletal muscle cells have been reported both in vitro and in vivo. Canonical FGF21 signaling occurs via its own receptors, FGFRs, and the co-receptor β-klotho, leading to the activation of downstream signaling pathways. β-klotho expression in the skeletal muscle is, however, very low [24,55,56], but might be sufficient to mediate FGF21 action [24,53]. Alternatively, β-klotho-independent FGF21 signaling pathways may exist in muscle, similar to that which occurs in adipose tissues [57]. In both human myotubes and mouse skeletal muscle, FGF21 was reported to induce the phosphorylation of FGFR, FRS (FGFR substrate) 2α and ERK1/2 [58]. In primary human myotubes and isolated adult mouse skeletal muscle, FGF21 enhances insulin-stimulated glucose uptake and GLUT1, but not GLUT4, protein expression at the plasma membrane [24]. In HFD fed mice, FGF21 increases insulin-stimulated AKT phosphorylation and glucose uptake in the skeletal muscle [22], while in human skeletal muscle myotubes, FGF21 prevents palmitate-induced insulin resistance [53]. Finally, in C2C12 myotubes, FGF21 was shown (i) to activate mTOR-p70s6 kinase-AKT-YY1-PGC1α signaling pathway leading to an enhanced mitochondrial function [59] and likely insulin sensitivity through PGC1α activation [60,61], and (ii) to increase fatty acid oxidation and potentially insulin sensitivity [59]. Consistent with these previous reports, our data further support a direct effect of FGF21 on insulin-induced AKT activation and insulin sensitivity of skeletal muscle cells.

High FGF21 expression and secretion are usually observed in obese mice, but with a concomitant downregulation of its cellular receptors in metabolic tissues [62], thus leading to the still debated concept that obesity is associated with a resistant state to the endogenous FGF21 action [63]. In this regard, the gene expression of FGFR1/2/3, the three major cellular receptors for FGF21, are in contrast upregulated in muscles of LPTENKO mice (Appendix A), further supporting the important contribution of FGF21 over-secretion in the muscle insulin hypersensitivity of these mice.

Differentiation and metabolic homeostasis of adipocytes is regulated by multiple mechanisms in pathophysiological conditions [64,65]. In this regard, our data highlighted a key role of the liver, which can modulate AT biogenesis and metabolism through the secretion of specific hormone-like mediators. Indeed, we found that CM collected from LPTENKO hepatocytes induced biologically significant changes in the expression of differentiation markers and key lipogenic factors in 3T3-L1 as compared to CM from CTL hepatocytes. A trend for similar changes in the expression of these differentiation and lipogenic factors was observed in the eWAT of LPTENKO mice, which contain much fewer lipids than the eWAT from CTL mice as we previously reported [34]. Several studies by other authors further reported similar changes in adipocytes markers in cases where adipogenesis is impaired. For example, KLF4 silencing in 3T3-L1 cells, which prevents lipid-droplet accumulation, is accompanied by a decrease of 22% for *Pparg* expression, 73% for adipsin and 54% for aP2 at the sixth day of differentiation [66]. Similarly, the inhibition of miR-140-5p, which blocks 3T3-L1 differentiation and lipid-droplet accumulation, is accompanied by a 60%, 50%, 75% and 35% decrease in the expression of *Pparg*, *Cebpa*, aP2, and adipsin, respectively [67]. We then identified three liver-secreted hepatokines, AHSG, ANGPTL4 and LECT2, which significantly impact the differentiation and lipid accumulation in adipocytes. Based on our data, it is probable that the effects of these hepatokines on adipocytes synergize, together with other pathological mechanisms, to promote fat mass expansion and obesity. Circulating levels of AHSG [68,69], ANGPTL4 [70,71] and LECT2 [14,72,73] were previously demonstrated to correlate with obesity, BMI or waist circumference in humans. Although these hepatokines are predominantly expressed by the liver, mouse and human adipocytes were also reported to produce them [74,75,76,77], therefore suggesting both endocrine and paracrine action of these factors on AT homeostasis.

Consistent with our data, AHSG (also known as fetuin-A) was shown to be required for the full differentiation of preadipose Ob17 cells into adipose-like cells [78]. In cultured primary adipocytes from dairy cows, AHSG was also reported to restrain lipolysis and to enhance lipogenesis via the upregulation of the rate-limiting lipogenic enzyme AGPAT2 [79]. Adiponectin expression was further found to be inhibited by AHSG in 3T3-L1 cells by mechanisms activating Wnt and Ras/MEK/ERK signaling and inhibiting PPARgamma, a nuclear receptor initiating adipogenesis [80,81]. Finally, the activation of SREBP-1C by AHSG, as was shown in HepG2 cells, could also constitute another molecular mechanism involved in the pro-lipogenic role of AHSG [82]. Interestingly, AHSG appears to promote AT inflammation and IR by acting as an endogenous ligand for TLR4 [12] and by fostering macrophage migration and M1 polarization in AT [83]. Based on these findings, PTEN-deficiency in hepatocytes, which lowers the production of AHSG, likely deprives the AT of an important stimulus for adipogenesis and inflammation, thereby improving systemic metabolic homeostasis in LPTENKO mice.

Conflicting data were reported regarding the role of ANGPTL4 (also known as fasting-induced adipose factor (FIAF), or the PPARgamma angiopoietin-related protein (PGAR)) in the lipid metabolism and adiposity. In agreement with our data, ANGPTL4 was reported to inhibit lipoprotein lipase (LPL) activity [84,85] and the specific silencing of ANGPTL4 either in white [86] or brown AT [87] resulted in enhanced LPL activity, AT lipolysis and fatty acids oxidation but a decreased fatty acids synthesis. The constitutive ANGPTL4 knockout in mice was further found to induce resistance to diet-induced obesity, suggesting a promoter role for ANGPTL4 in obesity [88]. Challenging these studies, ANGPTL4 knockout mice fed an obesogenic diet were also found to develop an increased body weight and visceral AT mass [89]. Transgenic mice mildly overexpressing ANGPTL4 further displayed and increased lipolysis and fatty acids oxidation, associated with a decreased food intake, body weight and adiposity [90,91]. The expression of ANGPTL4 was also found to be increased during 3T3-L1 [75,92] and SW872 [93,94] adipocyte differentiation, suggesting a promoter role for this hepatokine in adipogenesis. Given these discrepant results, additional studies are required to clarify the role of ANGPTL4, particularly in relation to the the central versus the peripheral effects of this factor in AT metabolism. Concerning the impact of ANGPTL4 on the liver, it was recently suggested that the specific depletion of ANGPTL4 in hepatocytes protects against diet-induced obesity and the development of hepatic steatosis [95], while hepatocytes exposure to dexamethasone triggers an ANGPTL4-dependent de novo lipogenesis and TG synthesis [96]. Based on these studies, while ANGPTL4 downregulation in the liver of LPTENKO mice could be expected to restrain steatosis, these mice still display an aberrant accumulation of lipids in hepatocytes probably because PTEN-deficiency overcomes this potential protective effect of ANGPTL4 depletion. However, our analyses still indicate that lowering the secreted levels of ANGPTL4 by hepatocytes in the culture medium of 3T3-L1 pre-adipocytes significantly prevents lipid accumulation in these cells, thus supporting a promoting role for ANGPTL4 in adipogenesis.

While the association of LECT2 overexpression with metabolic syndrome, fatty liver disease and cancer is well established [97,98,99,100,101,102], very few studies have investigated the role of this hepatokine in adipogenesis. Actually, two studies suggested that LECT2 triggers lipid accumulation in 3T3-L1 cells [13] and HepG2 cells [103] by activating the transcription factor SREBP-1C. The inhibition of lipid accumulation that we observed in 3T3-L1 cells exposed to CM from control hepatocytes depleted of LECT2 is consistent with these studies and supports a role for hepatic LECT2 downregulation in the decreased adiposity of LPTENKO mice. Whether this is related to a lack of SREBP-1C activation in 3T3-L1 cells remains to be investigated, but our data suggests that basal physiological levels of circulating LECT2 are necessary for efficient adipogenesis. Future studies are now required to better understand the molecular mechanisms driven by LECT2 in adipocytes and its role in both physiological and pathological conditions. 

In addition to hepatic steatosis, increased muscle insulin sensitivity and decreased adiposity, LPTENKO mice have also a low glycemia [34]. This raises the hypothesis that deregulated hepatokines may also contribute via autocrine/paracrine controls to an altered glucose consumption and/or release by LPTENKO hepatocytes. An increased glucose consumption by hepatocytes when PTEN is deleted or downregulated is supported by several pieces of evidence. We previously demonstrated that the mRNA expression of rate-limiting key enzymes in hepatic glycolysis (*Gck*, *Hk2*, *Pkm2*, *Pfkl*, *Pfklr*) were upregulated in the liver of LPTENKO mice [34]. Similarly, Stiles et al. reported that hepatic *Pten* deletion in mice induced an increased glycogen synthesis in the liver [104]. In vitro, PTEN deficiency in human HCC cell lines (HepG2 and Huh7) is further associated with an increased glucose consumption [105], whereas its overexpression decreased glucose uptake by the HHCC cells [106]. Taken together, these previous studies clearly indicate that PTEN knockdown or knockout in hepatocytes triggers an increased glucose utilization. Finally, the induction of PTEN expression in human glioma cells reduced by half their glucose consumption [107]. We can therefore not exclude an autocrine effect in vivo of the aberrantly secreted hepatokines on hepatocyte glucose metabolism with PTEN deficiency, but currently available data rather indicates an exacerbated PI3K activity in these processes. 

Based on our protein and mRNA analyses, FGF21 and AHSG are deregulated at the transcriptional level in the liver of LPTENKO mice, while the downregulation of ANGPTL4 and LECT2 protein expression appears to occur by means of post-transcriptional mechanisms. Investigating the precise molecular mechanisms regulating the expression and secretion of hepatokines, whose expression pattern is modulated in PTEN-deficient hepatocytes, was beyond the scope of our study. Our previous data suggest, however, that the improved metabolic phenotype of LPTENKO mice is not related to neither energy expenditure, which is unaltered in these mice as we previously reported [34], nor to liver hypertrophy and/or hyperplasia. Indeed, we did not observe hepatocyte hyperplasia in the liver of LPTENKO mice at the age of 4 months. Only hepatocyte hypertrophy associated with lipid accumulation in these cells (steatosis) seems to occur and to be responsible for the observed hepatomegaly [34,104]. These observations are further supported by publicly available transcriptomic data of hepatic tissues of 3- and 15-month old LPTENKO mice, showing that proliferation markers (KI67, cyclin A, D and E) are increased only in the tumoral liver tissues of 15-month old LPTENKO mice, and not in the liver of the 3-month old LPTENKO (Appendix A). Hepatocyte hypertrophy due to steatosis and hepatomegaly is a common feature of obesity-associated NAFLD. Indeed, hepatocyte hypertrophy/steatosis and hepatomegaly are also observed in genetic (e.g., leptin (ob/ob) or leptin receptor (db/db) deficient mice) and diet-induced mouse models of obesity and IR. However, to the best of our knowledge, only LPTENKO mice develop a fatty liver concomitantly with an improved metabolic phenotype of muscles and adipose tissues. All other genetic/diet-induced models of obesity/IR develop, on the contrary, metabolic disorders of the adipose tissues (e.g., hypertrophy, inflammation and IR) and muscles (e.g., ectopic fat accumulation and IR) together with hepatocytes hypertrophy/steatosis and hepatomegaly. It is therefore highly unlikely that liver hypertrophy, or steatosis per se, contribute to the observed improvement in muscles and adipose tissue metabolism in LPTENKO mice. We could, however, gain some insight into the role of the PI3K/PTEN signaling pathways in the transcriptional regulation of *Ahsg*, *Fgf21* and *Igfbp1*. PTEN deletion or downregulation triggers a longer and increased activity of AKT and ERK1/2 signaling in cells [108,109]. In both in vivo LIPTENKO mice and in vitro primary hepatocytes exposed to selective inhibitors, we observed a strong AKT, but not ERK1/2, signaling dependence of *Ahsg*, *Fgf21* and *Igfbp1* mRNAs expression. The inhibition of AMPK (by compound C), a major energy sensor, also significantly decreased the expression of *Fgf21* in MPH from LPTENKO mice. These results are in agreement with previous studies showing a control of FGF21 expression in the liver by PTEN [51] or in skeletal muscles by AKT signaling [110]. Of note, while the mRNA expression of *Angptl4* is not affected by PTEN deletion in the liver, we found that insulin downregulates its expression, as previously reported in 3T3-L1 cells [111]. This data suggests that cell-type specific PI3K-activated pathways independent of the PTEN status are implicated in the mRNA regulation of *Angptl4* reminiscent also of the complex regulatory mechanisms controlling *Igfbp2* expression. Indeed, inconsistent with the strong downregulation of *Igfbp2* mRNA that we observed in the liver of LPTENKO mice, PTEN was shown to inhibit IGFBP2 expression, while PI3K/AKT activation increases its expression in human brain and prostate cancer [112]. In addition, in preadipocytes, although PI3K inhibition prevented the production of IGFBP2, it was found to be completely insensitive to MAPK/mTOR inhibition and PTEN deficiency [113], whereas IGFBP2 expression was, in contrast, increased by the activation of the PI3K/AKT/mTOR pathway in MCF7 [114]. 

Although the potential therapeutic relevance of modulating the expression/activity of AHSG, ANGPTL4, LECT2 and FGF21 for treating obesity, IR and liver metabolic disorders is still unclear, targeting these hepatokines appears to be a promising strategy in addition to other currently used therapies. In primary hepatocytes, we found that classical antidiabetic and insulin-sensitizer drugs, e.g., metformin and rosiglitazone, as well as micronutrients known to improve metabolism and obesity, e.g., the isoflavone genistein, specifically modulate the expression of these hepatokines. Of note, all these drugs, which improve metabolic homeostasis, decreased the expression of AHSG which is consistent with previous studies reporting that metformin [115,116,117], thiazolidinediones [118,119] and phytoestrogens also inhibit the expression and secretion of AHSG or Fetuin B in various cells, animal models or patients. Regarding ANGPTL4, rosiglitazone and fenofibrate were reported to induce its expression in cardiomyocytes [120], adipocytes [75,121] and in human plasma [122,123], while genistein inhibits it in endothelial and blood mononuclear cells [124]. Although our data tend to confirm these regulatory mechanisms, we were unable to observe significantly different *Angptl4* expression in hepatocytes exposed to these drugs/nutrients. The fact that *Angptl4* expression is induced by anti-diabetic drugs further fuels the controversy about the role of ANGPTL4 in metabolic homeostasis as discussed earlier. Finally, *Lect2* expression was not found to be significantly sensitive to anti-diabetic drugs and phytoestrogens, while *Fgf21* expression was strongly stimulated by rosiglitazone and genistein in MPH, in agreement with previous studies in hepatocytes or mesenchymal stem cells [125,126]. However, we could not confirm the induction of *Fgf21* expression by metformin as previously suggested by other studies [127,128].

To the best of our knowledge, the clinical impact of targeting AHSG or LECT2 in patients suffering from metabolic disorders has not yet been assessed. For ANGPTL4, although clinical studies targeting this hepatokine are not currently available, a recent study in mice, where the *Angptl4* gene was specifically inhibited in the liver, highlighted its promising future therapeutic potential to improve metabolic homeostasis and prevent diet-induced obesity, without important adverse effects [95]. Only animal and clinical studies assessing FGF21-based therapies are well documented. Numerous studies using recombinant FGF21 proteins, FGFR-activating antibodies or long-acting FGF21 analogues have been performed to evaluate the relevance of such therapies to treat metabolic disorders in classical animal models [21,22,25,129,130,131,132,133,134,135,136], leading to promising results and no serious deleterious side effects. Several clinical trials using FGF21 analogues or new recombinant PEGylated human FGF21 have shown an improved dyslipidemia, body weight, adiponectin levels and fasting insulinemia, but unfortunately a low effect on the glycemic control [137,138,139]. Progress to improve, for example, the pharmacokinetic properties (e.g., bioavailability and delivery) of these FGF21-mimicking compounds are now required to improve the efficacy of these therapies against human obesity, IR and T2D.

PTEN is a potent tumour suppressor and its deletion in the liver leads to the development of hepatocellular carcinoma (HCC) and cholangiocarcinoma (CC) with ageing. Hepatokines can act also in an autocrine or paracrine manner on liver cells, therefore raising the question as to whether the deregulation of their expression in PTEN-deficient hepatocytes also contributes over time to hepatocarcinogenesis. The expression and secretion of several hepatokines increase in tumoural tissues such as HCC. For instance, serum and hepatic FGF21 expression increase in mouse and human HCC [51,140], but whether this hepatokine protects from, or fosters, hepatic carcinogenesis is still unclear. Indeed, while two studies have suggested that FGF21 protect from NASH to HCC transition and development [141,142], as well as from prostate and pancreatic cancer development [143,144], other studies support an oncogenic role of FGF21 in various other cancers, including thyroid [145], lung [146] and colorectal [147] cancers. Among the other hepatokines investigated in our study, high levels of AHSG [148,149,150], IGFBP1 [151,152,153] and IGFBP-2 [154,155,156] are found in the serum and HCC tissues of patients, whereas LECT2 expression is usually decreased and appears to behave as a tumour suppressor under the control of beta-catenin in the liver [101,157,158,159]. Finally, ANGPTL4 also appears to have a dual role in HCC, since its increased serum and hepatic levels are associated with an oncogenic activity [160,161], while at low expression/secretion levels ANGPTL4 acts as a tumour suppressor [162,163]. Analyses of publicly available transcriptomic dataset from HCC patients (Appendix A) further indicate that a significant proportion of HCC patients exhibit high levels of *FGF21*, but low levels of *AHSG*, *ANGPTL4*, *FETUB* and *LECT2* in HCC tissue as compared to matched non-tumoral tissue. Surprisingly, although such patterns of expression are associated with metabolic improvements in terms of IR and adiposity, they are corelated with a bad prognosis for HCC patients with the exception of FGF21, whose increased expression tends to be associated with a better prognosis (Appendix A). More in depth analyses are now required to evaluate the potential use of these hepatokines as biomarkers, or even therapeutic targets in liver cancers.

## 4. Materials and Methods

### 4.1. Cells, Antibodies, siRNAs, Primers, Kits, Reagents and Diets

Muscle, adipocyte and hepatic cell line cultures, as well as glucose uptake assays and lipid quantifications and stainings are described in detail in Appendix A and Methods. All antibodies, siRNAs, primers, kits, reagents and diets used during the study are listed in Appendix A and Methods.

### 4.2. Animals 

The liver-specific PTEN knockout AlbCre/Pten^lox/lox^ (LPTENKO) mice and their control littermate Pten^lox/lox^ (CTL) mice were previously described [164] and investigated at the age of 4 months. The liver-specific inducible PTEN knockout AlbCre-ERT2^Tg/+^/Pten^lox/lox^ (LIPTENKO) mice were produced by crossing AlbCre-ERT2^Tg/+^ with Pten^lox/lox^ mice. The 2-month old LIPTENKO mice were injected with tamoxifen to induce PTEN deletionm specifically in hepatocytes and were used 8 days or 3 months later. The 2-month old db/db and db/+ mice were purchased from Charles River Laboratories. Liver tissues from 2.5-month old ob/ob mice and their control were kindly obtained from Prof. Françoise Rohner-*Jeanrenaud* (CMU, University of Geneva). Mice fed an obesogenic high-fat enriched diet have been previously described [42]. Only males were used and all experimental procedures were conducted according to the Swiss guidelines for animal experimentation and were approved by the Geneva Health head office. Animals were housed at 23 °C, on a 12 h light-dark cycle (7 a.m.–7 p.m.). Standard food (SAFE-150 diet, SAFE, Augy, France) and water were available ad libitum. Mice were sacrificed using isoflurane anaesthesia followed by decapitation, and blood/tissues were rapidly collected and stored at −80 °C. 

### 4.3. Metabolomic Analysis

Bile acids levels in control and LPTENKO mouse plasma samples were determined using bile acids assay and amino acids, biogenic amines, acylcarnitines, (lyso)phosphatidylcholines, sphingomyelins and hexoses were measured using the Absolute *IDQ*^®^ p180 kit (Biocrates Life Sciences, Innsbruck, Austria).

### 4.4. Preparation of Conditioned Media from Primary Hepatocytes

Isolation and treatments of mouse primary hepatocytes are described in detail in Appendix A and Methods.

First, 5 h after plating, mouse primary hepatocytes (MPH) were washed twice with PBS and cultured for 24 h in deprivation medium. The conditioned media (CM) were then collected and filtered through 0.22 µM filters before use on differentiated C2C12 cells for 24 h or for the differentiation of 3T3-L1 cells for 8 days. Alternatively, 5 h after seeding, a part of MPH was transiently transfected with specific siRNA (20 nM) for 24 h using Viromer^®^ Blue following the manufacturer’s instructions. They were then washed twice with PBS and cultured for another 24 h in a deprivation medium in order to prepare the conditioned media. In a conditioned media fraction, the denaturation of secreted proteins was achieved by heating at 90 °C for 30 min followed by immediate storage on ice.

### 4.5. Proteomic Analysis of Conditioned Media by LC-MS/MS

*Sample preparation:* The protein concentration in conditioned media (prepared without BSA) was determined using the Bradford assay (Bio-Rad, Hercules, CA, USA). Proteomes (10 µg) were denatured with 6 M urea in 50 mM NH_4_HCO_3_ and then reduced with 10 mM TCEP for 30 min at room temperature followed by alkylation with 25 mM iodoacetamide for 30 min at room temperature in the dark. Samples were then diluted to 2 M urea with 50 mM NH_4_HCO_3_ and digested with trypsin for 12 h at 37 °C (Thermo Scientific, Waltham, MA, USA; 1 μL of 0.5 μg/μL) in the presence of 1 mM CaCl_2_. Finally, protein samples were acidified to a final concentration of 5% acetic acid, desalted using a self-packed C18 spin column and lyophilized. 

The *LC-MS/MS Analysis* and the *MS Data Analysis* were performed as previously detailed [35].

### 4.6. Protein Extraction and Western Blot

Proteins were extracted from cells or ex vivo liver tissues (30 µg) using ice-cold RIPA buffer (150 mM NaCl, 1% NP40, 0.1% SDS, 0.5% deoxycholicacide, 50 mM Tris-HCl pH 7.4, 0.5 M EDTA, 1 mM vanadate, 10 mM Naf, 1 mM PMSF and 1 tablet of *cOmplete*™ Protease Inhibitor Cocktail (Roche)). The supernatant was collected after centrifugation (10 min at 12,000× *g*) and the protein content was measured using a BCA protein assay with bovine serum albumin as standard (Pierce Biotechnology). Then, 10 µg of proteins was submitted to SDS-PAGE (5–20% gradient gels) and transferred on nitrocellulose membranes (Amersham, Dübendorf, Switzerland), which were then blocked in polyvinyl alcohol for 1 min. Next, proteins of interest were probed with specific primary antibodies overnight at 4 °C and, after several washings with TBS 0.1% Tween 20, with proper horseradish peroxidase-conjugated secondary antibodies for 1 h at room temperature. The signal was detected by PXi gel imaging system (Syngene, Cambridge, UK) and quantified with ImageJ software.

### 4.7. mRNA Extraction and RT-qPCR

RNAs were extracted from cells or liver pieces (30 µg) using TRIzol^®^ reagent (Life Technologies, Zug, Switzerland) according to the manufacturer’s instructions and dosed by Nanodrop 2000C (Thermo Scientific). A total of 1 µg of RNA was used for the reverse transcription using the “High Capacity RNA-to-cDNA” kit from Applied Biosystem. Quantitative PCR were performed using specific primers (Microsynth, Balgach, Switzerland), SYBR^®^ Select Master Mix and StepOne plus PCR system (Applied Biosystem, Waltham, MA, USA). The results were normalized with a housekeeping gene (*Ppia* (cyclophilin A-coding gene), *Rn18s* (18S ribosomal RNA-coding gene) or *Rps9*) and expressed as fold change relative to their control condition.

### 4.8. Bioinformatics Analysis 

All bioinformatic and in silico analyses are described in detail in Appendix A and Methods.

### 4.9. Statistical Analyses

Data are expressed as mean ± SEM of at least three independent in vitro experiments or with at least four mice per group. Statistical analyses were performed using GraphPad Prism 8 software. Outliers were removed from the analysis using RObust regression and OUTlier removal (ROUT) (Q = 1%). Differences between groups were tested for significance using a Student’s *t*-test (for 2 groups) or by one-way ANOVA (for multiple comparisons). *p-*value ≤ 0.05 was considered statistically significant. 

## Figures and Tables

**Figure 1 ijms-23-03959-f001:**
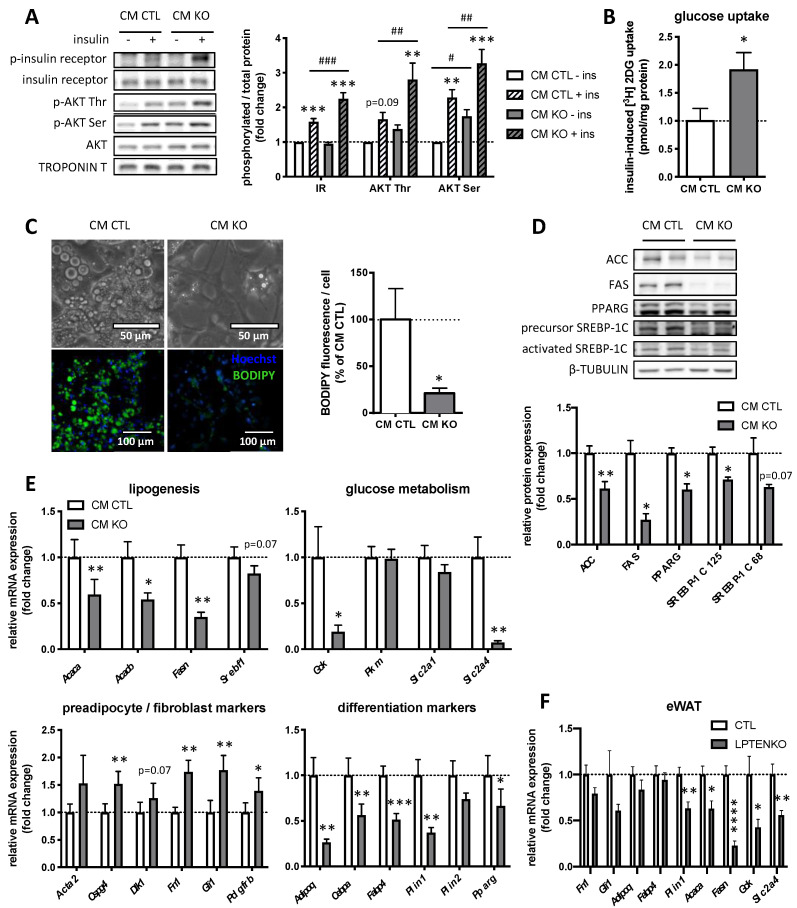
Regulation of muscle insulin sensitivity in C2C12 muscle cells and lipogenesis in 3T3-L1 adipocytes by conditioned media (CM) from LPTENKO hepatocytes. (**A**) Differentiated C2C12 cells were exposed for 24 h to conditioned media of primary hepatocytes isolated from 4 months old Pten^lox/lox^ (CM CTL) or LPTENKO (CM KO) mice and then treated or not with 10 nM insulin for 10 min. Representative Western blot analyses of phosphorylated insulin receptor, insulin receptor, p-AKT (threonine and serine) and AKT protein expression and quantifications. TROPONIN T was used as a gel loading control. Values are means ± SEM of 5–7 independent experiments. * indicates statistical significance of insulin treatment compared to respective control without insulin and # indicates statistical significance difference between CM CTL and KO. (**B**) Glucose uptake in C2C12 cells in response to insulin (100 nM for 20 min). Values are means ± SEM of 5 independent experiments. (**C**) Differentiation of 3T3-L1 preadipocytes was initiated in CM CTL or CM KO for 8 days. Bright-field and BODIPY/Hoechst-33342 staining pictures of lipid droplets and quantification. Values are means ± SEM of 3 independent experiments. (**D**) Representative Western blot analyses of acetyl-CoA carboxylase (ACC), fatty acid synthase (FAS), peroxisome proliferator-activated receptor gamma (PPARG) and sterol regulatory element-binding protein 1C (SREBP-1C) protein expression in 3T3-L1 cells and quantifications. β-TUBULIN was used as a gel loading control. Values are means ± SEM of 3–5 independent experiments. (**E**) Relative mRNA expression of key genes involved in lipogenesis and glucose metabolism, and of preadipocyte, fibroblast and differentiation markers, by RT-qPCR in 3T3-L1 cells. *Ppia* was used as reference gene to normalize the RT-qPCR analyses. Values are means ± SEM of 9 independent experiments. (**F**) Relative mRNA expression of preadipocyte, fibroblast and differentiation markers and of key genes involved in lipogenesis and glucose metabolism by RT-qPCR in epididymal adipose tissue (eWAT) of 4 months old control (CTL) and LPTENKO mice. *Rps9* was used as reference gene to normalize the RT-qPCR analyses. Values are means ± SEM of 7–9 animals. * or #: *p* ≤ 0.05, ** or ##: *p* ≤ 0.01, *** or ###: *p* ≤ 0.001, ****: *p* ≤ 0.0001 (one-way ANOVA for (**A**) and *t*-test for the others).

**Figure 2 ijms-23-03959-f002:**
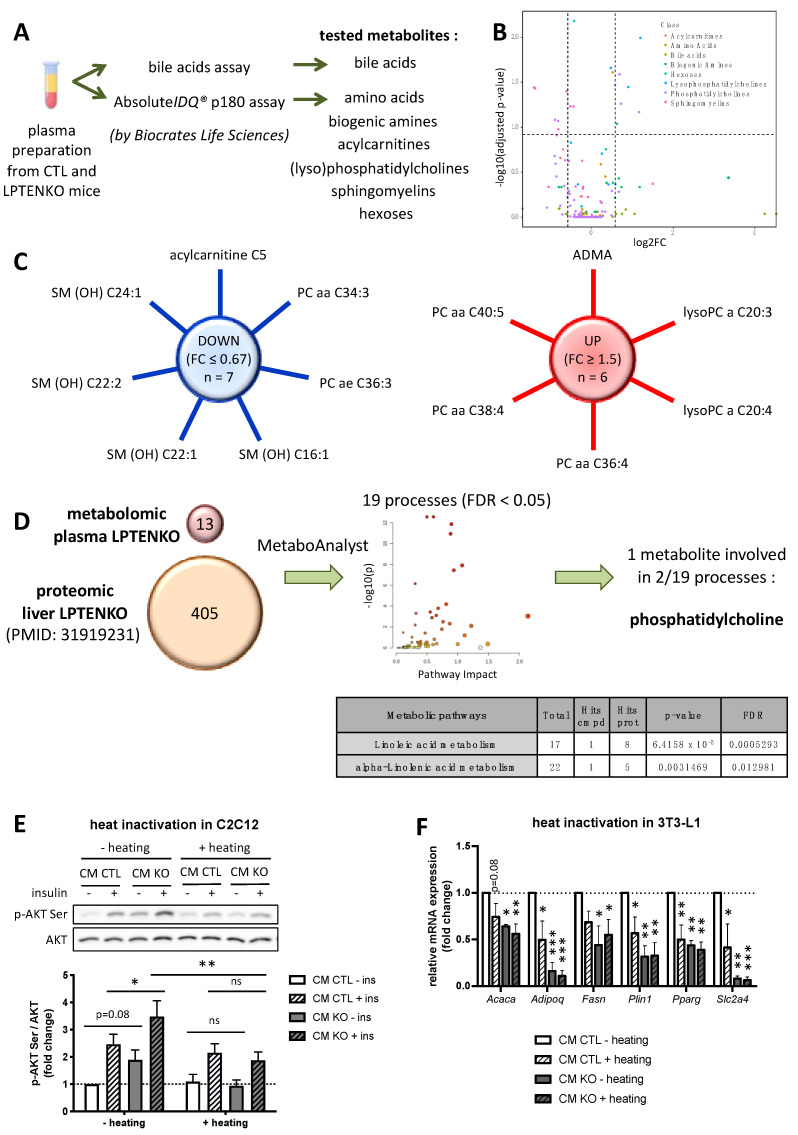
Metabolomic analyses of plasma from CTL and LPTENKO mice and thermosensitivity of CM. (**A**) Bile acids, amino acids, biogenic amines, acylcarnitines, lysophosphatidylcholines, phosphatidylcholines, sphingomyelins and hexoses were measured in plasma from control (CTL) and LPTENKO mice (Biocrates Life Sciences assays, 4 samples per group). (**B**) Volcano plot representation of the detected metabolites. Dashed lines represent fold change at 0.67 and 1.5 and adjusted *p*-value at 0.15 (*t*-test). (**C**) Sun diagrams summarizing the down- and upregulated metabolites in the LPTENKO plasma (fold change KO/CTL ≤ 0.67 and ≥1.5; adjusted *p*-value < 0.15). (**D**) Scheme illustrating the integration analysis performed on 02 February 2022 between the 13 deregulated metabolites of plasma from CTL vs. LPTENKO and the 405 deregulated proteins of liver from CTL vs. LPTENKO mice [35], using MetaboAnalyst 4.0. 19 processes were significantly enriched (false discovery rate (FDR) < 0.05). The table reporting pathways in which metabolites are involved is issued from the main table shown in Appendix A. (**E**) Representative Western blot analyses of phosphorylated AKT (serine) and AKT protein expression in differentiated C2C12 cells exposed to heat inactivated CM from CTL and LPTENKO isolated primary hepatocytes for 24 h and treated or not with insulin (10 nM, 10 min) and quantifications. Denaturation of proteins in the CM was achieved by heating at 90 °C for 30 min. Values are means ± SEM of 3 independent experiments. (**F**) Relative mRNA expression of lipogenic/differentiation markers by RT-qPCR in 3T3-L1 adipocytes exposed to heat inactivated CM from CTL and LPTENKO isolated primary hepatocytes. *Ppia* and *Rps9* were used as reference genes to normalize the RT-qPCR analyses. Denaturation of proteins in the CM was achieved by heating at 90 °C for 30 min. Values are means ± SEM of 3 independent experiments. *: *p* ≤ 0.05, **: *p* ≤ 0.01, ***: *p* ≤ 0.001 (one- and two-way ANOVA for (**E**,**F**), respectively).

**Figure 3 ijms-23-03959-f003:**
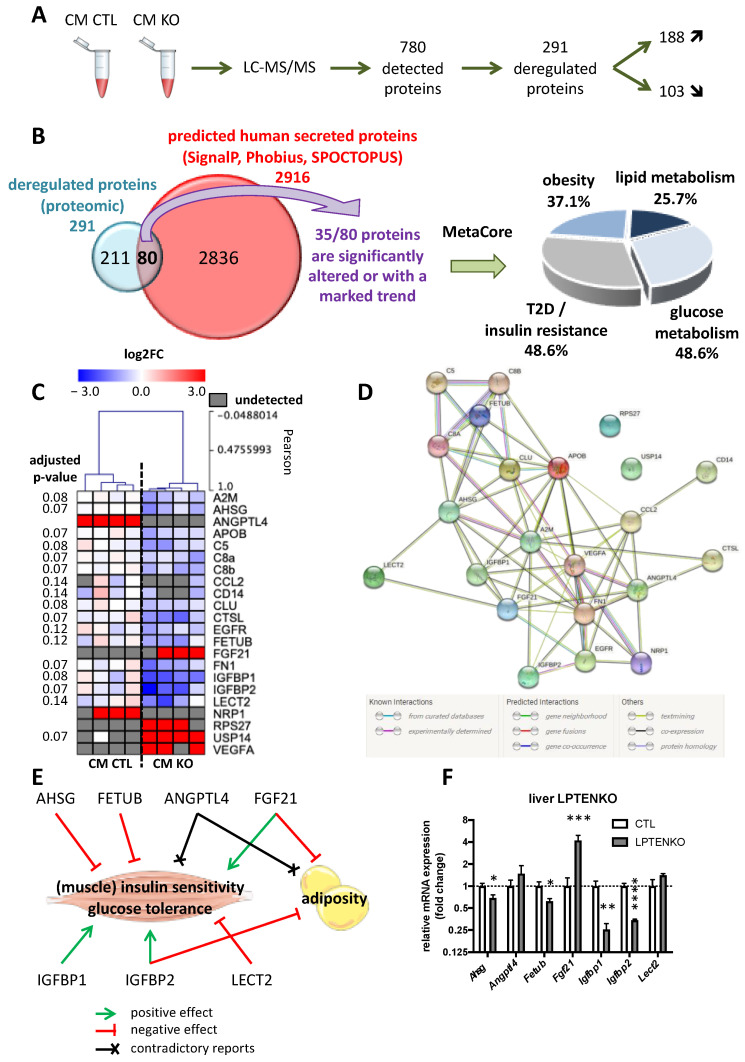
A network of hepatokines is deregulated in the liver of LPTENKO mice. (**A**) Differentially expressed proteins in conditioned media of primary hepatocytes isolated from 4 months old Pten^flox/flox^ (CM CTL) and LPTENKO (CM KO) submitted to liquid chromatography—mass spectrometry analysis. 4 samples per group. (**B**) Schematic representation of cross analysis of deregulated proteins in CM KO (fold change CTL/KO ≤ 0.25 and ≥1.33; adjusted *p*-value < 0.15) with predicted human secreted proteins based on three prediction methods (SignalP 4.0, Phobius and SPOCTOPUS, accessed on 14 November 2017). The list of identified candidates was then compared to a list of genes potentially involved in lipid and glucose metabolisms, obesity and/or type 2 diabetes (T2D) and insulin resistance (IR), obtained from Metacore database on 18 February 2020. (**C**) Heat map representation of the 22 candidate’s expression in CM CTL and CM KO. 4 samples per group. (**D**) Predicted or known protein–protein interactions and/or co-expressions from STRING database (accessed on 15 February 2022). (**E**) Sketch of the role of selected hepatokines on adiposity, insulin resistance and/or glucose tolerance in metabolic organs, based on literature (see Appendix A for the corresponding PMIDs). Green arrows indicate positive effects and red ones-negative. Black crosses indicate contradictory data. (**F**) Relative mRNA expression of hepatokines by RT-qPCR in the liver of 4month old control (CTL) and LPTENKO mice. *Ppia* was used as reference gene to normalize the RT-qPCR analyses. Values are means ± SEM of 8–9 mice per group. *: *p* ≤ 0.05, **: *p* ≤ 0.01, ***: *p* ≤ 0.001, ****: *p* ≤ 0.0001 (*t*-test).

**Figure 4 ijms-23-03959-f004:**
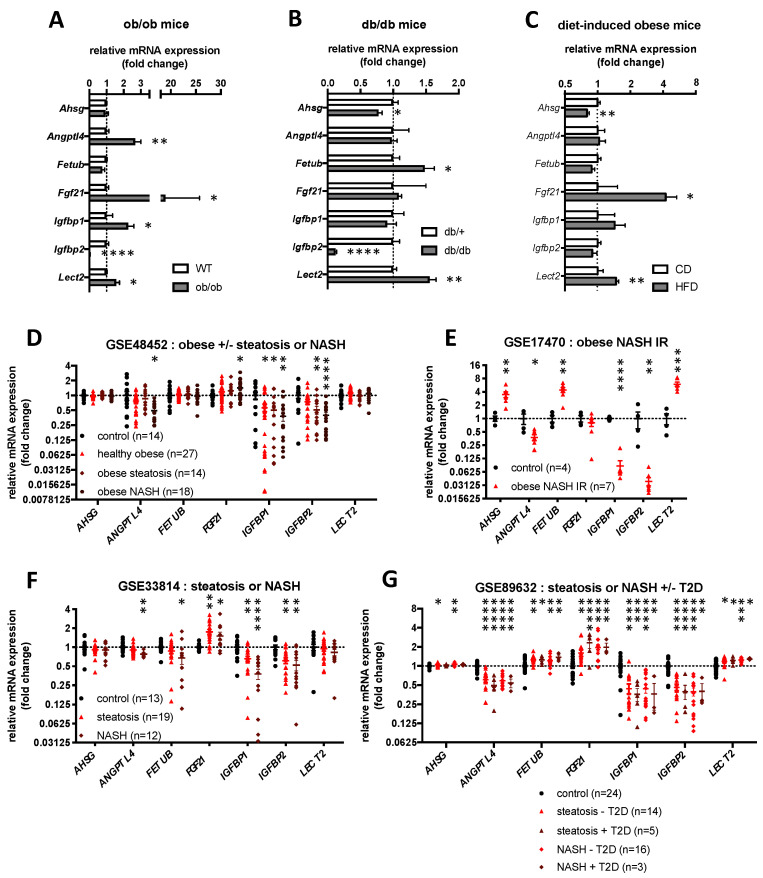
mRNA expression of hepatokines in the liver of mouse models and human with obesity, IR, NAFLD/NASH or type 2 diabetes. Relative mRNA expression of hepatokines by RT-qPCR in the liver of ob/ob mice (**A**), db/db mice (**B**) and mice fed for 18 weeks with a HFD (**C**). *Ppia* was used as reference gene to normalize the RT-qPCR analyses. Values are means ± SEM of 5–8 mice per group. (**D**–**G**) Hepatokines mRNA expression in the liver of several cohorts of obese, type 2 diabetes (T2D) and/or NAFLD/NASH patients obtained from public Gene Expression Omnibus (GEO) datasets. See Appendix A for details. *: *p* ≤ 0.05, **: *p* ≤ 0.01, ***: *p* ≤ 0.001, ****: *p* ≤ 0.0001 (*t*-test).

**Figure 5 ijms-23-03959-f005:**
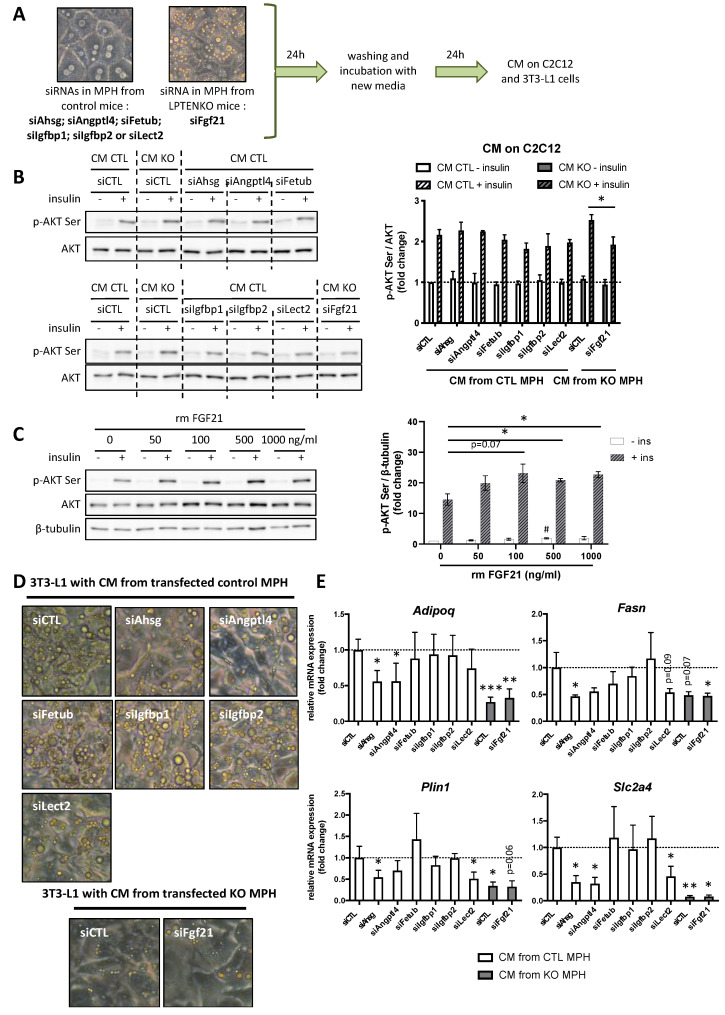
Identification of hepatokines improving muscle insulin sensitivity and triggering lipogenesis in adipocytes. (**A**) Overexpression of *Fgf21* in CM from LPTENKO MPH was decreased by specific siRNA to mimic expression of *Fgf21* in CTL CM. On the contrary expression of all other hepatokines were decreased by siRNA in CTL CM to mimic the expression in CM from LPTENKO MPH. 24 h after transfection with specific siRNAs, media were changed and conditioned media (CM) were collected after 24 h for incubation with differentiated C2C12 and 3T3-L1 cells. (**B**) Representative Western blot analyses of phosphorylated AKT (serine) and AKT protein expression in control and insulin-treated (10 nM for 10 min) C2C12 cells preincubated with conditioned media (CM) from siRNAs-transfected mouse primary hepatocytes (MPH) and quantifications. Values are means ± SEM of 3 independent experiments. (**C**) Representative Western blot analyses of phosphorylated AKT (serine), AKT and β-TUBULIN protein expression in control and insulin-treated (10 nM for 10 min) C2C12 cells preincubated with CM from control MPH containing different concentrations of recombinant mouse (rm) FGF21 and quantifications. Values are means ± SEM of 3 independent experiments. * indicates statistical significance between insulin-treated cells with rm FGF21 compared to insulin-treated cells without rm FGF21 and # indicates statistical significance between unstimulated cells with rm FGF21 compared to unstimulated cells without rm FGF21. (**D**) Bright-field pictures of 3T3-L1 adipocytes 8 days after the initiation of differentiation in CM from siRNAs-transfected mouse primary hepatocytes (MPH). (**E**) Relative mRNA expression of adiponectin (*Adipoq*), fatty acid synthase (*Fasn*), perilipin 1 (*Plin1*) and Glucose transporter type 4 (*Slc2a4*) by RT-qPCR in these 3T3-L1 adipocytes treated with conditioned media (CM) from siRNAs-transfected mouse primary hepatocytes (MPH). *Ppia* was used as reference gene to normalize the RT-qPCR analyses. Values are means ± SEM of 4 independent experiments. * or #: *p* ≤ 0.05, **: *p* ≤ 0.01, ***: *p* ≤ 0.001 (*t*-test).

**Figure 6 ijms-23-03959-f006:**
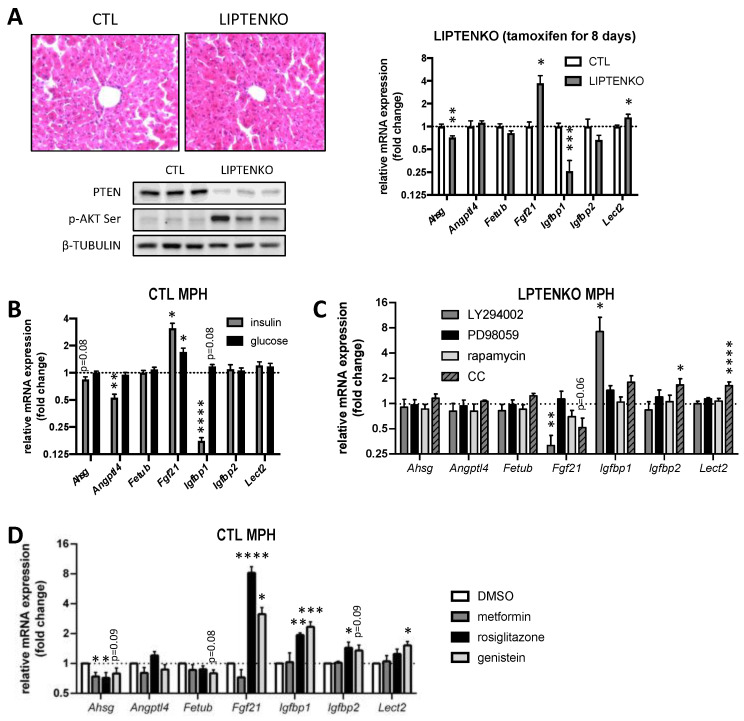
Regulation of hepatokines expression in hepatic tissues and isolated mouse primary hepatocytes. (**A**) Representative liver histology (H&E staining), representative Western blot analyses of PTEN and AKT phosphorylation (serine) and relative mRNA expression of hepatokines by RT-qPCR in the liver of liver-inducible PTENKO mice treated with tamoxifen for 8 days. β-TUBULIN was used as a gel loading marker and *Ppia* was used as reference gene to normalize the RT-qPCR analyses. Values are means ± SEM of 6 mice per group. (**B**) Relative mRNA expression of hepatokines by RT-qPCR in control mouse primary hepatocytes (MPH) treated with insulin (10 nM) or glucose (4.5 g/L) for 6 h. For each gene, the mean of respective control group is normalized to 1. *Ppia* was used as reference gene to normalize the RT-qPCR analyses. Values are means ± SEM of 4 independent experiments. (**C**) Relative mRNA expression of hepatokines by RT-qPCR in LPTENKO mouse primary hepatocytes treated with dimethylsulfoxide (DMSO, control), 20 μM LY294002 (PI3K inhibitors), 25 μM PD98059 (MEK inhibitor), 250 nM rapamycin (mTOR inhibitor) or 25 μM compound C (AMPK inhibitor) for 24 h. For each gene, the mean of control group (primary hepatocytes from LPTENKO mice treated with DMSO) is normalized to 1. *Ppia* and *Rps9* were used as reference genes to normalize the RT-qPCR analyses. Values are means ± SEM of 3–4 independent experiments. (**D**) Relative mRNA expression of hepatokines by RT-qPCR in control mouse primary hepatocytes (MPH) treated with dimethylsulfoxide (DMSO, control), 100 μM metformin, 20 μM rosiglitazone or 20 μM genistein for 24 h. *Ppia* and *Rn18s* were used as reference genes to normalize the RT-qPCR analyses. Values are means ± SEM of 4 independent experiments. *: *p* ≤ 0.05, **: *p* ≤ 0.01, ***: *p* ≤ 0.001, ****: *p* ≤ 0.0001 (*t*-test for (**A**,**B**), one-way ANOVA for the others).

## Data Availability

All the raw data supporting reported results can be found in Yareta (Data repository of Unige) using the following link: https://doi.org/10.26037/yareta:btlhnkmykrhkxdqtpv7srzwwhu. Raw data from our metabolomic and proteomic analyses are available in Appendix A. The lists of predicted human secreted proteins were retrieved from three different prediction algorithms accessed on 14 November 2017 (SignalP 4.0: http://www.cbs.dtu.dk/services/SignalP/, Phobius: https://phobius.sbc.su.se/ and SPOCTOPUS: http://octopus.cbr.su.se/). The lists of genes associated with lipid, glucose metabolism and T2D/insulin resistance were retrieved from the Metacore database accessed on 18 February 2020 (https://portal.genego.com/). Expression of hepatokines in transcriptomic datasets were obtained from the Gene Expression Omnibus database (https://www.ncbi.nlm.nih.gov/). Survival information in HCC patients were obtained from the Gepia database accessed on march 2020 (http://gepia.cancer-pku.cn/). Interactions between identified proteins were retrieved from the STRING database accessed on 15 February 2022 (https://string-db.org/). Data reporting the location and tissue specificity of identified proteins were retrieved from the human protein atlas (https://www.proteinatlas.org) and the human and mouse liver single cell atlas (http://www.livercellatlas.mvm.ed.ac.uk; https://tabula-muris.ds.czbiohub.org) accessed on 3 April 2020.

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
