# Peer review of "Hepatic PTEN Signaling Regulates Systemic Metabolic Homeostasis through Hepatokines-Mediated Liver-to-Peripheral Organs Crosstalk"

_ijms, 2022, doi:10.3390/ijms23073959_

Round 1

Reviewer 1 Report

The article "Hepatic PTEN signaling regulates systemic metabolic homeostasis through hepatokines-mediated liver-to-peripheral organs crosstalk" focuses on the identification of secreted factors regulating insulin sensitivity in muscles and the lipid metabolism in adipose tissue, which could be useful in obesity and insulin resistance treatment. The paper is well-conceived and the scientific results are consistent. 

Reviewer 2 Report

The manuscript by Berthou et al. “Hepatic PTEN signaling regulates systemic metabolic homeostasis through hepatokines-mediated liver-to-peripheral organs crosstalk” represents a comprehensive study aimed at identifying hepatocyte-derived molecules enabling muscle insulin sensitivity and lipodystrophy in a model of hepatocyte-specific PTEN deficiency. Through the employment of a number of in vitro and in vivo approaches coupled with highly sensitive analytical methodologies, the authors identified that protein factors, rather than specific metabolites, released by PTEN-deficient hepatocytes trigger an improved muscle insulin sensitivity and decreased adiposity. Moreover, FGF21 and the triad AHSG, ANGPTL4 and LECT2 have been identified as the key regulators of insulin sensitivity in muscle cells and in adipocytes biogenesis highlighting their potential use as targets for treating obesity and insulin resistance.

Reviewer 3 Report

In the presented manuscript, Berthou and colleagues aimed to identify hepatocyte secreted factors that could explain the increased muscle insulin sensitivity and decreased obesity in PTEN liver KO mice. The project is very ambitious and because of the very extensive work required (has to) remain somewhat superficial. In my opinion, the story would be better split into two different studies that evaluate the effects on either muscle or adipocyte differentiation. Such a more narrow focus would then allow to dedicate additional control experiments, such as titration series of inhibitors or supernatants, the use of different models etc, which would increase the confidence in the reported findings.

Some specific comments and suggestions:

  • Hepatic PTEN plays a central role in gating hepatocyte proliferation. Did the authors observe effects related to liver hypertrophy or hyperplasia and could metabolic effects be due to differences in energy expenditure?
  • Please provide all uncropped Western blots that were used for any of the column plots as Supplementary Material.
  • Figure 1C: The representative images and quantifications do not align. From the image it seems that the downregulation of lipid accumulation in CM KO is much higher than 50%.
  • The transcriptional changes of adipocytes in differentiation and dedifferentiation markers are rather miniscule. During differentiation, many of those factors change by 100-1000 fold and, thus, I am not convinced that the observed alterations of 20-50% have biological relevance.
  • Was glucose consumption altered in PTENKO hepatocytes?
  • For the metabolomic analysis (Figure 2), the significances should be corrected for multiple testing. As I understand, the threshold of p<0.08 refers to the nominal p-value.
  • Similarly, multiple testing correction should be applied to the proteomic data from Figure 3.
  • The authors show “mRNA analyses of these 7 hepatokines in the liver of 4-months old LPTENKO mice”. However, the proteomic analyses are done using conditioned medium from hepatocytes in culture. It would be good to analyze also the expression of the markers in the cultured hepatocytes after media conditioning to avoid indirect effects that stem from e.g. effects of PTEN on dedifferentiation in culture. This is particularly true as, again, the shown effects on hepatokine expression (Fig. 3F) are rather low.
  • I am not aware that direct effects of FGF21 on skeletal muscle signaling have been described. Could the authors please provide a potential mechanism that could explain direct action?
  • It would important if the authors could show effects of recombinant FGF21 on C2C12 cells in a concentration series. This would help to understand critical concentrations and whether/how these can be achieved in vivo. Similar experiments could be done with effects of AHSG, ANGPTL4 and LECT2 on 3T3-L1 differentiation.
  • Figure 6: The used concentrations of LY294002, PD98059 and compound C seem very high. What are the reported EC50 values for these compounds in the literature? Could these high concentrations result in off-target effects?
